# An Empirical Evaluation of Data Interoperability—A Case of the Disaster Management Sector in Uganda

**Allan Mazimwe** [1,*,†], **Imed Hammouda** [2,†] **and Anthony Gidudu** [1] 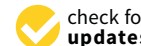

1   Department of Geomatics and Land Management, Makerere University, Kampala 00000, Uganda;
    agidudu@cedat.mak.ac.ug
2   Mediterranean Institute of Technology, South Mediterranean University, Tunis 1053, Tunisia;
    imed.hammouda@medtech.tn
*   Correspondence: allanmazimwe@cedat.mak.ac.ug
†   These authors contributed equally to this work.

**Abstract:** One of the grand challenges of disaster management is for stakeholders to be able to discover, access, integrate and analyze task-appropriate data together with their associated algorithms and work-flows. Even with a growing number of initiatives to publish data in the disaster management sector using open principles, integration and reuse are still difficult due to existing interoperability barriers within datasets. Several frameworks for assessing data interoperability exist but do not generate best practice solutions to existing barriers based on the assessment they use. In this study, we assess interoperability for datasets in the disaster management sector in Uganda and identify generic solutions to interoperability challenges in the context of disaster management. Semi-structured interviews and focus group discussions were used to collect qualitative data from sector stakeholders in Uganda. Data interoperability was measured to provide an understanding of interoperability in the sector. Interoperability maturity is measured using qualitative methods, while data compatibility metrics are computed from identifiers in the RDF-triple model. Results indicate high syntactic and technical interoperability maturity for data in the sector. On the contrary, there exists considerable semantic and legal interoperability barriers that hinder data integration and reuse in the sector. A mapping of the interoperability challenges in the disaster management sector to solutions reveals a potential to reuse established patterns for managing data interoperability. These include; the federated pattern, linked data patterns, broadcast pattern, rights and policy harmonization patterns, dissemination and awareness pattern, ontology design patterns among others. Thus a systematic approach to combining patterns is critical to managing data interoperability barriers among actors in the disaster management ecosystem.

**Keywords:** disaster; hazard; FAIR; data interoperability; patterns

## 1. Introduction

Disasters cause severe losses to communities, the effects of which are felt by the most vulnerable in communities. In order to cope with impacts of hazard events, organizations require resources and capabilities of other stakeholder institutions thus making effective collaboration and inter-operation essential. To make full use of existing data in the disaster management sector, the disaster community needs to adopt mechanisms that ensure resource findability, accessibility, interoperability, reuse and promote the growth of open data initiatives. Even with growing open data initiatives, integration and reuse of existing data by sector stakeholders is hindered by existing interoperability barriers. The barriers in sharing and coordinating information before and after hazard events often leads to

a breakdown in prevention, preparedness and response to extreme hazard events [1]. Thus lack of interoperability for disaster/hazard related data is in itself an indicator of lack of resilience. Wilkinson et al. [2,3] discuss guiding principles to make data in repositories Findable, Accessible, Interoperable and Reusable (FAIR) and associated metrics for evaluating compliance. In the disaster domain, [4] have developed a framework for assessing interoperability as well as other domain interoperability frameworks listed by [5] can be reused in the disaster management sector.

While these frameworks provide a way of measuring interoperability, they lack approaches for identifying and development of best practice solutions to interoperability challenges based on interoperability measurement. A combination of interoperability concepts and patterns have been proposed as a way of overcoming the interoperability barriers [6,7]. Patterns provide reusable proven best practice solutions for recurring interoperability problems in a given context [8]. However, there still exists a gap in pattern demand and availability to enable domain experts to construct high quality interoperability solutions without requiring expert knowledge in interoperability.

Therefore, the contribution of this paper is: First, to provide an understanding of interoperability maturity and compatibility between existing data sets in the disaster management sector in Uganda. Second, from interoperability assessments, we identify potentially reusable interoperability patterns—also known as interoperability best practices. In this paper, we define interoperability patterns as typical solutions to commonly reoccurring interoperability problems within in the context of disaster management. These patterns can be systematically combined to manage interoperability problems in the disaster management sector in Uganda. The rest of the paper is structured as follows: Related work is presented in Section 2, materials and methods in Section 3 while results and discussions in Sections 4 and 5 respectively. Section 6 contains the conclusions and future work.

## 2. Related Work

Currently, there exists is a strategic shift in disaster risk management with actors placing more attention on proactive disaster preparedness and prevention as opposed to reactive measures [9]. Some of the proactive measures include rigorous risk assessments, enhanced early warning systems, awareness and coordination of actors. Early warning typically constitute multiple actors (such as national sectors, private sector, civil society and communities) playing different yet complementary roles in disaster risk management [10,11]. The success of an enhanced early warning system typically depends on the ability of multiple actors to share data/ information (i.e., data about past events, real-time streams etc.) [12,13].

### 2.1. Disaster Data Interoperability

Vast amounts of hazard/disaster-related data existing today are typically dispersed geographically and owned by various entities making it difficult to access and reuse for disaster management. As such, Open Data initiatives aiming at increasing the availability of machine-readable data provided under an open license and facilitate data reuse have been advanced [14–16]. The FAIR principles defined by [2] provide a guide for ensuring such open disaster related data is findable, accessible, interoperable and reusable (FAIR) by machines/humans to support disaster management functions and so become as valuable as is possible [17]. The concept of interoperability serves to enable creation, exchange and consumption of data with explicit, shared expectation of context and meaning [18]. Different frameworks have been advanced to reconcile several perspectives on the notion of data interoperability [19]. These frameworks include the IMI [20], EIF [21] and LCIM [22] to mention but a few. After reviewing these frameworks, Rezaei et al. [23] categorises data interoperability into several distinct levels. These include: syntactic, semantic, technical and organizational interoperability. To ensure that data providers publish a machine-readable open data in an interoperable way, Barners-lee [24,25] introduces a five-star rating for data that relates to the levels of interoperability defined by Rezaei et al. [23].

## 2.2. Interoperability Assessment

Leal et al. [5] classify interoperability assessment types used by different frameworks into three categories. The first assessment type is the potentiality assessment that evaluates whether a system is mature enough to overcome existing interoperability barriers. Examples of such an assessment include [26] used in Uganda and [4] in the disaster domain. The second type is the compatibility assessment that evaluates the current state of data to identify potential conflicts that may cause interoperability problems as exemplified in [22,27,28]. The last assessment type evaluates the quality of implementing an interoperable solution at run-time. While these frameworks provide a way to measure interoperability, Leal et al. [5] conclude that they lack approaches for identifying best practices necessary to improve interoperability based on assessment results.

## 2.3. Interoperability Patterns

To solve interoperability challenges, different authors [6,7,23] propose a combination of interoperability concepts and patterns (also known as information supply chain patterns). The concept of patterns emerges from software engineering [29], data modeling [30] and knowledge engineering [31] among others to interoperability. To overcome technical interoperability barriers, several patterns have been developed as best practice solutions. Examples of application domains where interoperability patterns have been applied to manage technical barriers include the Internet of Things (IoT) [32], enterprise information systems exchange (a catalog of patterns can be found here: http://project-interoperability.github.io/exchange-patterns/) [33] and health domain [7]. In the semantic ecosystem, Gangemi [34] proposes ontology design patterns (ODPs) as common strategies for implementing semantic interoperability [8]. To solve organization interoperability barriers [6,7] identify several patterns. While the prospects for using interoperability patterns to manage interoperability barriers are enormous, there still exists a gap between their demand and availability [35]. Thus, there is a need to develop a critical amount of reusable patterns for managing disaster/ hazard related data interoperability.

## 3. Materials and Methods

In this study, we use the case of Uganda that experiences several disasters such as famine, floods, earthquakes, conflict, landslides and epidemics. With each hazard, there exists a lead institution (See the list of lead institutions in the Uganda disaster preparedness and management policy that can be downloaded from here: https://reliefweb.int/report/uganda/national-policy-disaster-preparedness-and-management) responsible for monitoring, assessing and reporting risk levels. For the institutions to achieve their mandates, they need to share, integrate and reuse data from other co-opted stakeholders. Through the disaster management sector working group, the Office of the Prime Minister (OPM) coordinates all stakeholders (data publishers and users) to promote integrated monitoring, reporting and actions to minimize hazard impacts. As a matter of fact, the current study investigates interoperability issues that affect the sharing and reuse of data among actors in the disaster management sector. In terms of theoretical scope, we investigate the data interoperability concern wherein two types of data are considered for the study (i.e., raw data and derived data like meteorological data obtained after processing raw sensor data). Therefore, this paper mainly addresses two research questions (RQ).

- *RQ1: To what extent is disaster/hazard-related data interoperable in Uganda?*
  Knowledge of data interoperability conflicts and maturity level are needed to guide the mapping of appropriate solutions to barriers in sharing/reusing data in the disaster management sector.
- *RQ2: Are there emerging interoperability patterns for managing data interoperability barriers in the disaster sector?*
  Pattern instances revealed through the mapping of local use case interoperability problems in

the sector to generic solutions are critical for stakeholders to implement appropriate solutions with ease.

To answer RQ1, two assessment types (i.e., the maturity and compatibility assessment) are used to explore data interoperability in the disaster sector as shown in Figure 1. For the maturity test, data are collected using semi-structured interviews from seven data points representing disaster sector stakeholders (see Table A3) that generate and reuse data in the sector. To gain more insight into interoperability issues in the disaster sector, data from members of the disaster sector working group was collected through the formulation of four Focused Group Discussions (FGD) constituting a total of 16 data users from 12 stakeholder institutions (see description of institutions in Table A4). The disaster sector working group is an inter-agency disaster risk reduction platform that constitutes the following categories of actors; lead Ministries, UN agencies, NGOs, CSOs, researchers and other relevant stakeholders as defined in the disaster policy. In this study, we endeavor to cover all actors by selecting at least a data point in each category to guarantee good representation. Thus a total of 15 institutions were selected to participate in the interviews and FGD based on purposive sampling. This represents about 24.6% of the total number of sector stakeholders (The derived percentage is based on 61 institutions included in the DRR platform/sector mailing list of 20th July 2018). The compatibility assessment utilizes structured data obtained during prioritization in FGDs. However, disaster sector data is both derived and raw (i.e., requires pre-processing for use as input in assessment models e.g., extracting indicator data from imagery). To avoid researcher bias that could affect the internal validity of the study, we do not pre-process any raw data into derived data but rather use existing data.

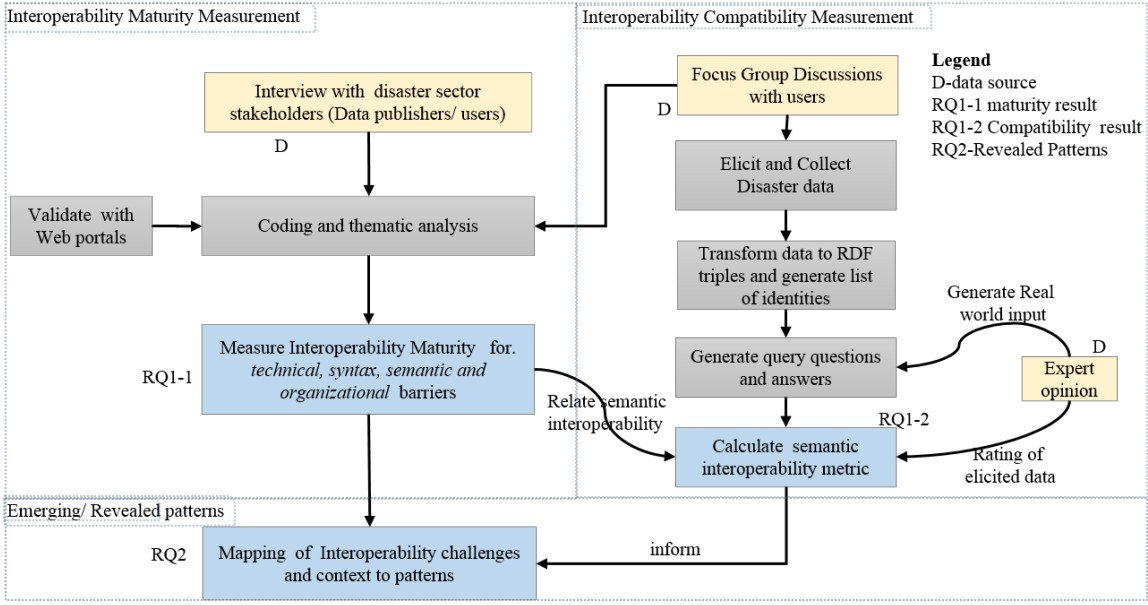

**Figure 1.** General flow of methodology.

Qualitative data collected from interviews and FGD were transcribed into text and later coded in atlas-Ti. The codes were then grouped into code groups and analyzed using content thematic analysis. To indirectly measure the interoperability maturity (i.e., the ability to overcome existing interoperability barriers), metrics in Table 1 were developed and classified according to data interoperability barriers presented by [23].

**Table 1.** Interoperability maturity metrics.

| Technical Interoperability | |
|---|---|
| T1: | Data is published in any format and accessed via a common protocol such as Http URL |

| Syntactic Interoperability | |
|---|---|
| S1: | Data is available as structured data e.g., shapefile, excel instead of scanned image |
| S2: | Data is available in a machine readable non proprietary format e.g., CSV, XML, GML, RDF, OWL etc |

| Semantic Interoperability | |
|---|---|
| Sem1: | Use URI to denote concepts for ensuring unique identification |
| Sem2: | Data is available with vocabularies that are linked |

| Organization Interoperability | |
|---|---|
| O1: | Licence attached to data |
| O2: | Data can be accessed freely |
| O3: | Have institutional arrangements to share data |
| O4: | Uses a data policy that is binding to other stakeholders in the sector(one that is not only institutional) |

To generate metrics that are in line with recommendations of the FAIR framework, we combine FAIR metrics [3] with the 5-star rating for data [25]/ linked open data [24] as illustrated in Table A1. The decision of the final metric is based on a generalization of FAIR interoperability metrics and data rating concepts. Where no FAIR interoperability metrics exist, 5-star rating metrics are selected taking into consideration their compliance to findability, accessibility and reusability metrics of the FAIR framework e.g. in metric T1. A numerical measure of maturity is generated by assigning compliance value codes (where 0 means false, 1 for True and 0.5 for partially) based on metrics in Table 1. Finally, the maturity indicator is computed by averaging compliance values.

To measure interoperability compatibility, test datasets (research tools and data used can be found here https://github.com/mazimweal/mazimweal.github.io) are compared using the method in Colpaert et al. [27]. This method is best suited for evaluating semantic inter-linkedness between RDF datasets. However, it also provides a good experimentation on the semantic inter-linkedness and similarity of concepts in tabular datasets using simple RDF syntax.

Consider two datasets A and B, (seen in Figure 2) being assessed for compatibility. Four scenarios regarding sharing of identifiers and concepts are possible (see Figure 2). Below is an explanation of implications for scenarios in Figure 2 on interoperability.

- *Scenario 1:* Does not affect interoperability since neither identifiers nor concepts are shared.
- *Scenario 2:* Denotes **true ID** matches where identifiers increase interoperability since they represent different concepts that are shared by datasets.
- *Scenario 3:* Two datasets share an identifier for which the concepts do not match. Therefore, it denotes a **false ID** match that negatively affects interoperability.
- *Scenario 4:* Also negatively affects interoperability since concepts match in the two data sets yet the identifiers do not match.

In this method, a pair of identifiers (IDs) shared by two datasets is evaluated to determine whether they represent the same concept. Therefore, the interoperability calculation will only consider scenarios 2, 3, 4 and exclude 1.

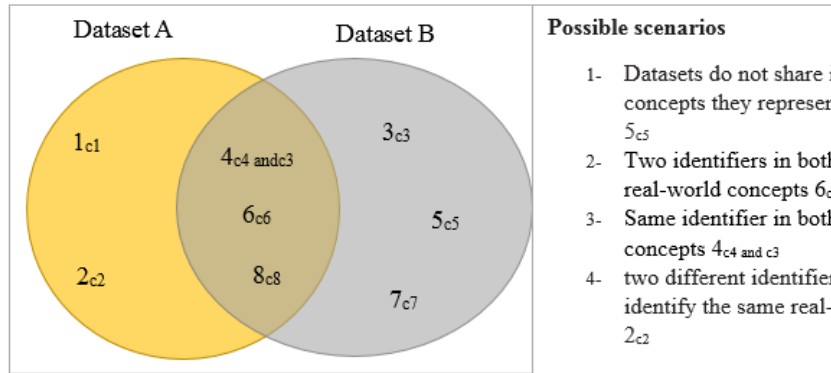

**Figure 2.** Dataset comparison (adopted from Colpaert et al. [27]).

The data (as illustrated in Figure 3) are transformed into the RDF format from which a list of identifiers is generated (see example in Figure 4). We then combine all identifiers from participating datasets into one file (totaling to 149 IDs).

| Serial | Event | District | Location | Date | Cause | Source | Death | Crops damaged |
|--------|-------|----------|----------|------|-------|--------|-------|---------------|
| 7027 | STORM | Amuria | Senior quarters | 2016 | Heavy rain | OPM | 6 | 10 |

a) Disaster Impact data (Source: Desinventar extract)

| Id | Disaster type | Branch | Region | Location | Response date | Response time | Population | Population targeted | Comments |
|----|---------------|--------|--------|----------|---------------|---------------|------------|---------------------|----------|
| 7062 | STORM | bubulo | Mbale | bududa | 2016 | 11.00pm | 50671 | 396 | damage to crops and households |

b) Disaster response data- (Uganda Red Cross Society)

**Figure 3.** Data extracts.

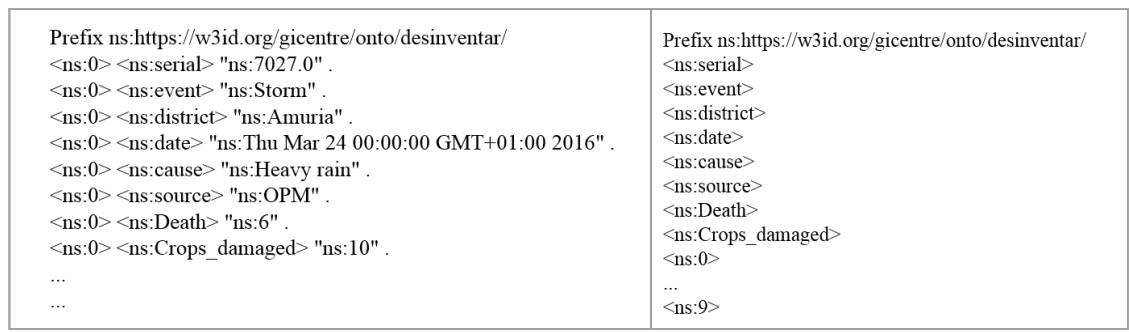

a) N3 format

b) List of IDs generated

**Figure 4.** NTriple (N3) and ID generation from datasets in Figure 3.

Using Algorithm (Figure 5a), every pair of identifiers in any two datasets is compared to generate a list of questions of the form $< id1 >?unknown < id2 >$ (i.e., from 149 IDs, we generate 149 $\times$ 148 = 22,052 questions). The list of IDs are then compared for concept similarity by replacing the ?*unknown* predicate with the *iiop:SameAs* or *iiop:notSameAs* depending on real-world inputs crowd-sourced from domain experts. The relationship between two IDs is denoted by *iiop:SameAs* if concepts are the same and *iiop:notSameAs* (the identifier interoperability vocabulary can be found here; http://semweb.mmlab.be/ns/iiop#) when different in two datasets as shown below.

```
Prefix ns1: <https://w3id.org/gicentre/onto#>
@prefix iiop: <http://semweb.mmlab.be/ns/iiop#>
<ns1:desinventar/Event>iiop:SameAs <ns1:response/Disaster_type>
<ns1:desinventar/Location>iiop:SameAs <ns1:response/Location>
<ns1:desinventar/Serial>iiop:SameAs <ns1:response/id>
```

The EYE reasoning server (find reasoning examples here https://n3.restdesc.org/n3/) [36] was then used to reason through the questions thus generating a list of answers (also known as interoperability statements) based on query Figure 5b. EYE is a semantic web reasoning engine that supports Euler paths. Through interoperability with Cwm via NTriple (N3), EYE provides a mechanism to query check, transform and filter information (See query in Figure 5b).

| | |
|---|---|
| **while** read id1 **; do {**<br>  **while** read id2 **; do {**<br>    **[[** $id1 **!=** $id2 **]] &&** echo "$id1 ?unknown $id2 . " ;<br>  **}** done **<** ids.txt **;**<br>**}** done **<** ids.txt **>** questions.nt | @prefix iiop: <http://semweb.mmlab.be/ns/iiop#> .<br>{ ?z iiop:sameAs ?x . } => { ?x iiop:sameAs ?z . } .<br>{ ?z iiop:notSameAs ?x . } => { ?x iiop:notSameAs ?z . } .<br>{ ?z iiop:sameAs ?x . ?x iiop:sameAs ?y . }<br>=><br>{ ?z iiop:sameAs ?y . } .<br>{ ?z iiop:notSameAs ?x . ?x iiop:sameAs ?y . }<br>=><br>{ ?z iiop:notSameAs ?y . } . |
| a) Generating list of questions | b) Query to generate list of answers |

**Figure 5.** Computing identifier interoperability [27].

To measure semantic interoperability, the open DesInventar dataset is used as a reference for computing the interoperability ratio in Equation (1). DesInventar (click this link to access the Desinventar dataset: https://www.desinventar.net/DesInventar/profiletab.jsp?countrycode=uga) is an open disaster management information system managed by the mandated disaster management institution (OPM). The system enables collection, documentation and analysis of data about losses caused by disasters associated with natural hazards. Thus it helps to understand disaster trends and their impacts for better prevention, mitigation and preparedness among communities.

$$Identifier Interoperability = \frac{\rho_{identifiers}}{\rho_{real-world}} \tag{1}$$

Consider a pair of datasets (see Figure 2), the $\rho_{real-world}$ relevance in Equation (1) is equal to the occurrence of consolidated identifiers in a joined dataset. While $\rho_{identifier}$ relevance is calculated similarly as the real-world relevance except that the strings in the identifiers and the respective concepts they represent need to match. In other words, the $\rho_{identifier}$ is the number of triples in a joined dataset with matching true positives (see scenario 2 in Figure 2). Based on expert opinion, the elicited datasets used in the compatibility assessment are rated to determine how easy it is to merge them with the reference dataset. These ratings are compared with interoperability metrics from Equation (1).

To answer RQ2, the resulting challenges from qualitative data analysis are matched with the problem part of pattern templates [37] to generate a list of relevant patterns in the disaster domain as illustrated in Figure 6. The solution space constitutes of generic patterns applied in other contexts other than disaster data interoperability. These were derived from literature and existing catalogs. The decision on the final pattern selected also takes into consideration the results from data interoperability assessment in the sector.

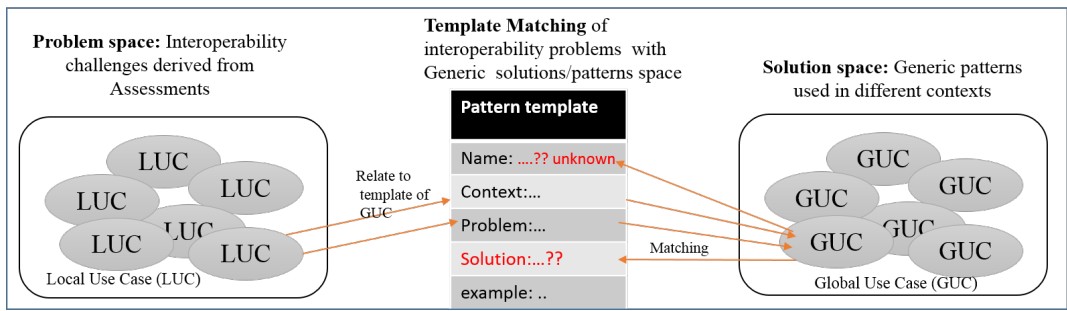

**Figure 6.** Problem and solution matching resulting into reusable patterns, i.e., "interoperability patterns".

## 4. Results

### 4.1. To What Extent Is Disaster/Hazard-Related Data Interoperable among Stakeholders?

This section presents results of interoperability maturity level and compatibility measures for disaster data in Uganda. The results are organized based on the four interoperability levels defined by [23].

#### 4.1.1. Technical Interoperability

Technical interoperability is reached when data/information can be shared and accessed using common standard protocols. For instance, a good indicator of technical interoperability is the ability to access disaster data via Http/URL and download it fully (see metric T1 in Table 2). All interviewed respondents manage web portals with downloadable disaster data/information except for one as shown in Table 2. This implies that technical interoperability can fully be achieved in the disaster sector in Uganda. In addition, sharing of disaster data is also achieved through email and flash-drives.

#### 4.1.2. Syntactic Interoperability

From Table A2, all participating organization have unstructured data (e.g., pdf), structured data (S1) and machine readable data (S2). Data are encoded in different formats depending on intent for data/information use. For instance, the intent of keeping disaster data as spreadsheets best suits analysis whereas shape file format suits mapping. The only open machine readable format used is the CSV, and that no institution uses RDF and XML machine readable formats for sharing data. Interview and FGDs indicated that the respondents had no working knowledge of XML and RDF which are also semantic markup languages signaling a lack of capacity in linked data/semantic technology among data publishers and users (see Section 4.1.4). However, since there existed both structured and machine readable data all of which the serializations are common among all institutions, we can conclude from Table 2 that at 93%, syntactic interoperability for data in the disaster management sector is high.

**Table 2.** Interoperability maturity (see Tables 1 and A2).

| Interoperability Level | Technical | Syntax | | Semantic | | Organisation | | | |
|---|---|---|---|---|---|---|---|---|---|
| Metric | T1 | S1 | S2 | Sem1 | Sem2 | O1 | O2 | O3 | O4 |
| Respondent 1 | 1 | 1 | 0 | 0 | 0 | 0 | 0.5 | 1 | 0 |
| Respondent 2 | 1 | 1 | 1 | 0 | 0 | 0 | 1 | 1 | 0 |
| Respondent 3 | 1 | 1 | 1 | 0 | 0 | 1 | 1 | 1 | 0 |
| Respondent 4 | 1 | 1 | 1 | 0 | 0 | 1 | 1 | 1 | 0 |
| Respondent 5 | 1 | 1 | 1 | 0 | 0 | 0 | 0.5 | 1 | 0 |
| Respondent 6 | 1 | 1 | 1 | 0 | 0 | 0 | 0.5 | 1 | 0 |
| Respondent 7 | 0 | 1 | 1 | 0 | 0 | 0 | 0.5 | 1 | 0 |
| **Total** | 6 | 7 | 6 | 0 | 0 | 2 | 5 | 7 | 0 |
| **Average metric** | 86% | 100% | 86% | 0% | 0% | 14% | 71% | 100% | 0% |
| Maturity metric | 86% | 93% | | 0% | | 46% | | | |

### 4.1.3. Organization/Legal Interoperability

A logical indicator for measuring legal interoperability is whether data has an open license such that the machine can understand what the reuse rights are. Generally, disaster sector stakeholders have no licenses attached to data (O1 in Table 2) making it hard to determine the openness of the data available. However, most of this data can freely be accessed, downloaded and reused (O2). In some instances, only derived data are available rather than the raw data e.g., climatic data. Inter-institutional arrangements among users and publishers exist, as illustrated in O3 to enable sharing of disaster related data. The data publishers asserted that they have institutional policies on how data should be accessed. This makes it hard for data users to access the data because each institution has its own policy as illustrated in the quote below.

> *Respondent 4;* "We know data is there but obtaining it is a problem. In most cases you need to use informal means to access it"

To solve these organizational interoperability barriers, there is a need to encourage license definition for data as well as harmonization of policies for maintaining and accessing data in the sector.

### 4.1.4. Semantic Interoperability

From Table 2, all respondents (data providers) have not used base URI for their data (Sem1) and neither do they have linked vocabularies (Sem2) describing concepts in their dataset. However, from qualitative results, semantic issues still exist while utilizing disaster/hazard-related data as illustrated in the quotes below.

> *Respondent 1:* "The climate domain uses difficult terminology that is often understood differently in other domains or by the common man for example the concepts *above average rainfall, late start, dry spell* are understood differently for agricultural domain and yet for layman it could also mean something else"

> *Focus group discussion 2;* "Since the desinventar database is populated by different volunteers, we are not sure whether all volunteers will conceptualize events for reporting purposes in the same way. For instance, there is no clear reporting on terms like rains, heavy rains, storms, windstorms, hailstorms."

The quotes above imply that data users are aware that semantic issues exist when reusing disaster/hazard-related data. From Section 4.1.2, we note that there is a lack of capacity in using semantic markup. Therefore, the absence of implementations to manage semantic interoperability issues in data can be explained by the lack of capacity among data producers. An important indicator of whether data can be semantically integrated is through comparison of identifiers in different data sets to check for concepts that are similar in the real world. Upon experimenting with the RDF model, results in Table 3 generally indicate a high interoperability identifier ratio except for the land cover dataset. Since the land cover dataset does not share identifiers and concepts with any other dataset and vice-versa, it's interoperability identifier value is zero. Thus has no impact on the interoperability of other datasets (see scenario 1 in Figure 2). Therefore, the results in Table 3 indicate that data in the disaster management sector is highly semantically inter-linked.

**Table 3.** Interoperability identifier ratio with the DesInventar dataset as reference.

| Dataset | Relevance of String Matching IDs | Relevance of Real-World Concepts | Interoperability Identifier | Remark |
|---|---|---|---|---|
| Landcover | 0 | 0 | 0% | low |
| Rainfall data | 20 | 20 | 100% | high |
| Emdat_impact data | 209 | 300 | 69.7% | medium |
| Risk data | 32 | 32 | 100% | high |
| Response data | 270 | 330 | 81.8% | high |
| Poverty data | 38 | 38 | 100% | high |

However, the semantic issue presented by respondent 1 requires a more expressive language to *infer* extreme weather events/conditions from rainfall data that can not be captured by simple RDF syntax used in the experiment above. While results in Table 3 suggest that linked data could improve semantic inter-linkedness between sector datasets, a more expressive way for representing explicit but also shared conceptualizations (i.e., through use of formal ontologies in OWL-DL) could be explored. Organizing sector data with ontologies requires an understanding of the complex and abstract ontology modeling notions expressed in OWL-DL. However, disaster domain experts are not often familiar with these constructs and will thus need a steep learning curve [38]. To mitigate the issue of lack of technical capacity in semantic technologies(as expressed in Section 4.1.2), best practice solutions (ontology design patterns and tools) can be adopted to make ontology modeling easy and intuitive enough for domain experts to develop ontologies for organising sector data efficiently and correctly without the help of Knowledge Engineers.

## 4.2. Are There Emerging Patterns for Managing Data Interoperability in the Disaster Management Sector?

Table 4 presents patterns (a description of revealed patterns is available here: https://github.com/mazimweal/mazimweal.github.io) obtained by mapping interoperability challenges with the problem part of the pattern template in the context of disaster management. Stakeholders in the disaster management sector share both raw and derived data. To help integrate and reuse data, respondents recommended the inclusion of metadata with pre-processing techniques. Also, respondents noted that shared data often has gaps, duplication and erroneous entries. To solve these data issues, respondents recommend best practices that will not only fill sector data gaps by enrichment but also ensure data validation with related datasets from other stakeholders. Linked data is a suitable technology for increasing data interoperability as explored by results in Table 3 not only by uniquely interlinking concepts but also through validation and enrichment of sector data with other sources. Therefore, several linked data patterns (a description of the patterns for modeling publishing and consuming of linked data are available at http://patterns.dataincubator.org/book/) for publishing and consuming linked data emerge. These include the "Follow Your Nose pattern" to find additional relevant data, "Progressive Enrichment pattern" for data quality and "Missing Isn't Broken pattern" to handle messy and incomplete data.

A technical interoperability barrier identified in the sector is the inability to access data scattered in multiple disaster/hazard related repositories. The federated query pattern provides a solution to enable access to multiple data stores with a single query. Furthermore, disaster/early warning managers expressed need for a solution that can send geolocated alert or notification to stakeholders across multiple areas to communicate details of specific hazard events. The broadcast pattern would be instrumental in solving such a problem.

From Section 4.1.3, there is a difficulty in finding and accessing data from publishers as organizations have different policies and regulations for maintaining/accessing disaster-related data. Best practice solutions from respondents show the need for organizations to advertise their data. The political will of the government to invest in data is critical. Besides, mandated government institutions ought to take lead in the formulation and implementation of policies supporting the growth of open data initiatives and harmonization of data sharing practices. Thus the following patterns emerge for reuse: dissemination and awareness pattern, policy development and harmonization patterns (e.g. rights pattern, private policy pattern, policy override and conflict mediation patterns), coordination and implementation patterns. These patterns are synonymous with pattern instances described in a different context by Nehta [7].

**Table 4.** Emerging interoperability patterns.

| Level | Interoperability Challenges | Recommended Solution | Emerging Patterns |
|---|---|---|---|
| General | 1-Missing meta data<br>2-Missing, incomplete, and erroneous datasets<br>3-Data is not up to date<br>4-No/ immature standards<br>5-Duplication of data collection efforts. | 1-Metadata documentation portal<br>2-Enriching and linking disaster /hazard related data to fill gaps and validation of data<br>3-Develop standards for sector data | 1-Linked data patterns e.g.,<br>-Follow Your Nose pattern to find additional relevant data<br>-Progressive Enrichment pattern for data quality<br>-Missing Isn't Broken pattern to handle messy and incomplete data |
| Technical | 6-Data is not easily retrievable because it is scattered | 4-Centralized access to repositories with linked documents | 2-Federated queries pattern |
| Technical | 7-Absence of directed alerts, notifications based sharing | 8-Geo-tagged alerts, and notifications<br>9-Embrace changing open architectures | 3-Broadcast pattern |
| Syntax | 8-Lack of capacity in semantic mark-ups and technologies | 10-Semantic systems that hide their complexity behind an easy to use and intuitive interface | – |
| Semantic | 9-Terminology is not standard across datasets to be integrated | 11-Unified vocabulary across domains e.g., glossary. However, the vocabulary should be expressive across different domains | 4-Content Ontology design patterns (ODPs) |
| Organisation | 10-Data is not easily accessed and retrieved but it is available in different institutions<br>11-Limited funding for data collection and management in the sector - | 12-Mandated institutions should take lead in policy formulation<br>13-Develop policies that Open up government data (i.e. embrace Open Government Data (OGD) principles)<br>14-Government should invest in data management<br>15-Create awareness and disseminate available disaster resources<br>16-organizations should add licenses to data | 5-Coordination and implementation pattern<br>6-Dissemination<br>7-privacy policy pattern<br>8-Policy harmonization(i.e. policy override and conflict mediation patterns)<br>9-Rights pattern |

To resolve semantic barriers identified in Section 4.1.4, respondents proposed a unified vocabulary across domains and institutions (see Table 4). Given that institutions maintain data based on different institutional policies as illustrated in Table 2, a bottom-up approach to ontology engineering that does not necessitate policy harmonization is preferable. As such, we identify content ontology design patterns for organizing data in the disaster management sector. Content ODPs allow institutions to align themselves to the interoperability infrastructure/global pattern schema on a gradual basis depending on the organizational capacity (see Section 4.1.4) and policies.

## 5. Discussion

### 5.1. Disaster Data Interoperability Evaluation

From the results above, there exists a high level of maturity for technical and syntactic interoperability for data in the sector. Data users can only access machine-readable CSV format (for instance, one can easily download disaster impact data from DesInventar database as CSV) implying that data is limited to the 3rd star of open data rating by BarnesLee [25]. On the contrary, semantic and organization interoperability is still a challenge in the sector. Since the study uses FAIR interoperability metrics for interoperability assessment, the maturity results show that disaster/hazard related data are far from fulfilling FAIR recommendations. In this paper, we explore a way of resolving interoperability issues through a mapping to best practice solutions. Therefore, future work could also focus on holistically exploring the FAIRness of disaster/hazard related data by further evaluating its findability, accessibility and reusability.

The identifier interoperability provides a mechanism to reason about the semantics of concepts in datasets based on the RDF data model. Results from compatibility measurement suggest that using the linked data can improve interoperability. However, users in the disaster management sector often use both structured, unstructured data (like pdf) and raw imagery that can be easily transformed using well-established procedures. A big challenge to this study is obtaining only the structured data available in the disaster sector to enable examination of conflicts in data. To address this problem, we triangulate data sources and interoperability assessment types to gain a bigger picture of data interoperability in the disaster sector. This allows demonstration of the fact that RDF alone is not

sufficient to capture complex semantic notions (e.g. inferring extreme weather events/conditions from rainfall data/thresholds). This implies that a more expressive mechanism is required to capture such semantic constraints of data such as OWL-DL. However, there is a lack of technical capacity in semantic markup technologies such as XML/RDF/OWL. Thus, future work can focus on furthering work on intuitive RDF interfaces (such as RDF data generators https://rdforms.org/#!start.md), and ODPs [39] to support managing of data by users/ disaster domain experts.

From a legal perspective, clearly defined license (open/proprietary) increases interoperability for both humans and machine. In Table 2, most disaster-related data are freely available but have no license attached. Thus machines will not explicitly understand data reuse rights even when data are seemingly open. Even with inter-institutional collaborations, data interoperability in the sector will remain a challenge since institutions maintain data based on different policies/regulations. The mandated hazard lead government institutions have a role to develop policies that will open and harmonize data management among stakeholders. Since the role of policy development is outside the purview of a typical user/stakeholder in the disaster sector, there is a need to develop interoperable disaster management solutions that take into consideration changing policies and related constraints.

*5.2. Interoperability Patterns*

The interoperability assessment (in Section 4.1) together with elicited challenges enable identification of suitable interoperability best practices for managing data in the disaster management sector. Patterns identified can be grouped using different classification systems such as categorization by interoperability barrier (see Table 4), granularity (e.g., federated and broadcast patterns are more coarse than ODPs) etc. Construction of high-quality interoperability solutions by domain experts without expert knowledge entails a systematic effort to combine patterns such as those in Table 4. Figure 7 illustrates the concept of pattern combination in the context of a multi-hazard early warning system. This suggests that a catalog of patterns and tools for combining these patterns must be readily accessible. While patterns identified in Table 4 are specific to interoperability issues in Uganda's disaster management sector, future work could also focus on the identification of new patterns at the different classification levels in a broader disaster community. The choice of pattern instances used in a composition is largely dependent on organizational policies and business processes. For instance, the selection of content ODPs in Section 4.2 to manage semantic barriers takes into consideration the fact that multi-hazard indicator data is managed based on different organizational policies. Therefore, applying patterns without reflecting on organizational policies and processes can lead to incorrectly modeled interoperability solutions.

At a coarse level(in this study, coarse level patterns are similar to architectural patterns), Figure 7 combines the federated and broadcast patterns to enable query multi-hazard data from different actors in the sector and generate alerts and notifications based on the queries. At a finer scale, willing institutions organise and publish hazard-related linked data using ontology design patterns as illustrated in [40]. Publishing data using ODPs allows for continuous connection to the unified global ontology pattern schema that is implicitly glued together with a foundational ontology. This linked hazard data can further be enriched and validated with related data on the web using linked-data patterns. A federated data query can then be posted onto data sources of various actors in the disaster management sector using interfaces such as that defined by [41]. The reasoning possibilities of ontologies facilitate the deduction of new facts during federated data queries that are critical for early warning.

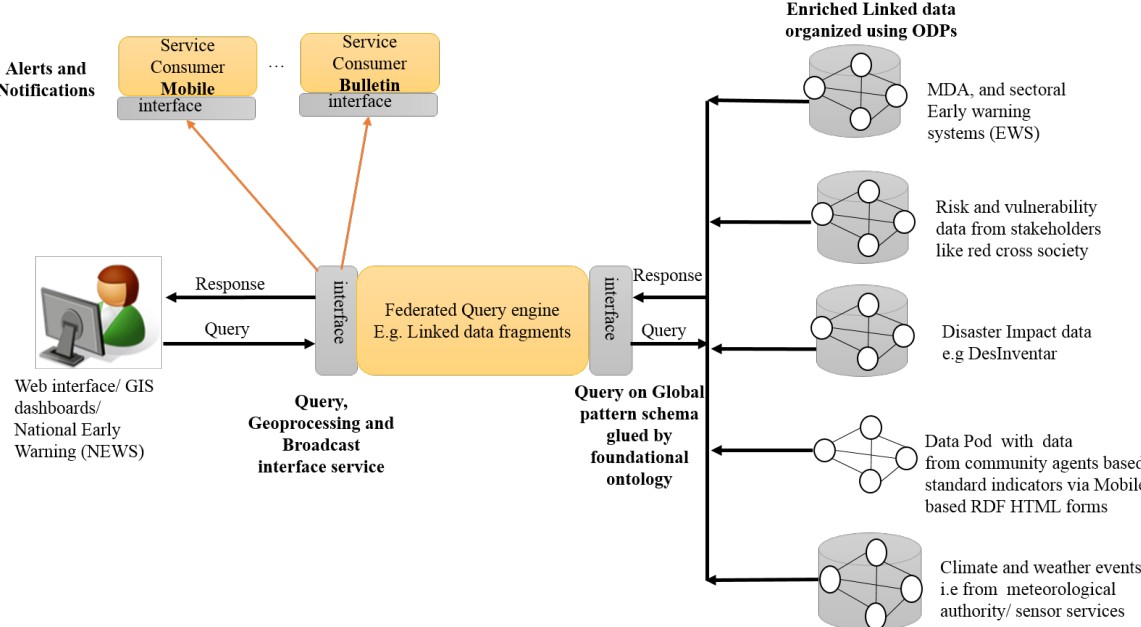

**Figure 7.** Illustrative example of a modular pattern composition in an early warning system.

Nonetheless, federated queries on linked data require that disaster data users and publishers are knowledgeable in the query technology in use e.g., SPARQL or GraphQL-LD. In Table 4, respondents recommended semantic systems with interfaces that hide such complexity. To overcome this challenge, future work could explore the use of natural language by data users/novices to make federated queries to decentralized repositories where semantic issues have been treated with ODPs. The work of [42,43] is useful to guide this conversation in the disaster management sector.

## 6. Conclusions

The success of disaster risk management efforts depends on the ability of multiple stakeholders to integrate and reuse existing disaster data. In this paper, we evaluate the extent to which data from multiple stakeholders in the disaster management sector is interoperable and identify patterns for reuse while solving interoperability challenges in the domain. Results indicate high technical and syntactic interoperability maturity for data in the sector. On the contrary, the maturity of semantic and legal interoperability is still low. The maturity results show that data in the disaster sector is far from fulfilling FAIR interoperability recommendations. Data users and publishers in the sector have a clear understanding of semantic issues existing in the datasets but lack the technical capacity to implement semantic technologies. Moreover, there also exists gaps in published data that are not easily accessed from multiple repositories. To solve these issues, we identify patterns that will enable disaster domain experts to construct highly interoperable solutions without the help of interoperability experts. Pattern identification takes into consideration the results of interoperability assessments. Pattern reuse requires a systematic effort to combine them to solve existing data interoperability barriers in the sector. The choice of patterns used will typically depend on organizational policies and business processes. In this study, we use a multi-hazard early warning solution to illustrate modular interoperability pattern reuse based on the assessment. The architecture of a multi-hazard early warning system will typically constitute the federated and broadcast patterns at a coarse level and linked data published using ODPs at a finer level. Publishing disaster data as linked data using ODPs is critical for data enrichment, validation and handling of semantic issues while taking into consideration the technical capacity and organizational constraints. Linked data from multiple publishers can, therefore, be integrated using federated queries. Thus future work could focus on the use of natural language for ODP based linked data query answering over federated repositories. This will enable data users

to access data without technical competence in semantic technologies. More so, future work could consider identification of more interoperability patterns at different classification levels. For instance, at a finer level, researchers could identify specific ontology design patterns that can be reused to publish data in the disaster management sector. This study assesses interoperability and provides best practices solutions to data interoperability challenges based interoperability metrics of the FAIR framework. Future work could also focus on holistically exploring the FAIRness of disaster data/vocabularies in Uganda not only by further evaluating its Findability, Accessibility, and Reusability but also proposing ways to increase its FAIRness.

**Abbreviations**

The following abbreviations are used in this manuscript:

| | |
|---|---|
| CSOs | Civil Society Organisations |
| CSV | Comma separated variable format |
| Cwm | Cwm (pronounced coom) is a general-purpose data processor for the semantic web |
| DRR | Disaster Risk Reduction |
| EWS | Early Warning Systems |
| EIF | European Interoperability Framework |
| EYE | Euler Yet another proof Engine |
| FAIR | Findable, Accessible, Interoperable and Reusable |
| FGDs | Focused Group Discussions |
| GraphQL-LD | GraphQL Query Language for linked data |
| HTML | Hypertext Markup Language |
| HTTP | Hypertext Transfer Protocol |
| IMI | Information Modeling and Interoperability |
| LCIM | Level of Conceptual Interoperability Model |
| MDA | Ministries, Departments and Agencies |
| NEWS | National Early Warning System |
| NGOs | Non-Governmental Organisations |
| NOAA | National Oceanic and Atmospheric Administration |
| ODPs | Ontology Design Patterns |
| OSM | Open Street Map |
| OWL_DL | Web Ontology Language -Description Logic profile |
| RDF | Resource description framework |
| SPARQL | Language for querying linked data |
| UN | United Nations |
| UNMA | Uganda National Meteorological Authority |
| URI | Uniform Resource Identifier |
| USGS | United States Geological Survey (Context: data explorer) |
| XML | Extensible Markup Language |

**Author Contributions:** Conceptualization, I.H., and A.M.; methodology, A.M. and I.H.; software, A.M., data curation, A.M. and A.G.; writing—original draft preparation, A.M.; writing—review and editing, I.H., A.G. and A.M.; supervision, I.H. and A.G.

**Funding:** This work is supported by the Swedish International Development Agency (SIDA) funding to Makerere University, Kampala in partnership with Chalmers and Gothenburg University, Sweden under the BRIGHT project 317.

**Acknowledgments:** We acknowledge support from Pieter Colpaert with regard to ideas on the FAIR framework and compatibility assessment.

**Conflicts of Interest:** The authors declare no conflict of interest.

## Appendix A. Interoperability Maturity

*Appendix A.1. Interoperability Maturity Metrics*

A full list of FAIR Metrics and corresponding descriptions can be found at https://github.com/FAIRMetrics/Metrics/tree/master/MaturityIndicators/Gen1.

**Table A1.** Defining metrics for interoperability maturity assessment.

| Rezaei et al. (2014) | 5 Star Data Rating (Berners-Lee 2006) | 5 Stars of Linked Data Vocabulary Use (Janowicz et al. 2014) | FAIR Metrics | Metrics Derived for Study |
|---|---|---|---|---|
| Technical | 1st Star:Make your stuff available on the Web (whatever format) under an open license | | FM-A1.1: Access Protocol is open, free, and universally implementable | T1: Data is published in any format and accessed via a common protocol such as Http URL |
| Syntactic | 2nd Star: make it available as structured data (e.g., Excel instead of image scan of a table) | | | S1: Data is available as structured data e.g., excel instead of a scanned image |
| | 3rd Star: Make it available in a non-proprietary open format | 2nd Star: The information is available as machine-readable explicit axiomatization of the vocabulary. E.g OWL, RDF W3C standards | FM-RI: Meets Community Standards (meta)data meet domain-relevant community standards<br><br>FM-I1: Use a Knowledge Representation Language-(meta)data use a formal, accessible, shared, and broadly applicable language for knowledge representation | S2: Data is available in a machine-readable non-proprietary format e.g., CSV, XML, RDF, OWL, XML, GML etc |

**Table A1.** *Cont.*

| Rezaei et al. (2014) | 5 Star Data Rating (Berners-Lee 2006) | 5 Stars of Linked Data Vocabulary Use (Janowicz et al. 2014) | FAIR Metrics | Metrics Derived for Study |
|---|---|---|---|---|
| Semantic | 4th Star: use URIs to denote things, so that people can point at stuff | 1st Star: Linked Data without any vocabulary | | Sem 1: Use URI to denote concepts to enable unique identification |
| | 5th Star: link your data to other data to provide context | 3rd Star: The vocabulary is linked to other vocabularies<br><br>4th Star: Metadata about the vocabulary is available<br><br>5th Star: The vocabulary is linked to by other vocabularies (note: this is the reverse of 3rd star) | FM-I1: Use a Knowledge Representation Language: (meta)data use a formal, accessible, shared, and broadly applicable language for knowledge representation<br><br>FM-I2: Use FAIR Vocabularies: IRIs representing the vocabularies used for (meta)data<br><br>FM-I3: Use Qualified References: (meta)data include qualified references to other (meta)data | Sem2: Data is available with vocabularies that are linked via unique and qualified references |

**Table A1.** *Cont.*

| Rezaei et al. (2014) | 5 Star Data Rating (Berners-Lee 2006) | 5 Stars of Linked Data Vocabulary Use (Janowicz et al. 2014) | FAIR Metrics | Metrics Derived for Study |
|---|---|---|---|---|
| Organisation | Ist Star: Make your stuff available on the Web (whatever format) under an open license | | FM-R1.1: Accessible Usage License: (meta)data are released with a clear and accessible data usage license | O1: License attached to data<br><br>O2: Data is accessed freely (but may does not have explicit license but it is openly accessible *Observed in interview data*) |
| | | | FM-F1B: Identifier persistence-Whether there is a policy that describes what the provider will do in the event an identifier scheme becomes deprecated. | O4: Uses a data policy that is binding to all institutions |
| | | | | O3: Presence of institutional arrangements (*From interview data*) |

*Appendix A.2. Summary of Data Used for Measuring Maturity*

**Table A2.** Qualitative measurement of disaster data interoperability.

| Respondent | Syntax | Technical | Semantic | Organisation |
|---|---|---|---|---|
| Respondent 1 | Excel/CSV, html, ppt, docx, pdf, shapefile | Email, websites-URL, Database | No URI and<br><br>No linked data | Data License: not sure<br>Data is open: Partially<br>Policy: institutional<br>Institutional arrangements:yes |
| Respondent 2 | Excel/CSV, shapefile, pdf, html | Email, websites-URL, Flash Drive | No URI and<br><br>No linked data | Data License: No<br>Data is open: yes<br>Nature of Policy: institutional<br>Institutional arrangements |
| Respondent 3 | html, shapefile, OSM | Website-URL | No URI and<br><br>No linked data | Data License: yes<br>Data is open: yes<br>Nature of Policy: institutional<br>Institutional arrangements:Yes |
| Respondent 4 | Excel/CSV, html, ppt, docx, pdf, shapefile | Website, URL | No URI and<br><br>No linked data | Data License: yes<br>Data is open: yes<br>Nature of Policy: institutional<br>Institutional arrangements:Yes |
| Respondent 5 | Excel/CSV, html, ppt, docx, pdf | Database, Website-URL, email | No URI and<br><br>No linked data | Data License: No<br>Data is open: yes<br>Nature of Policy: institutional<br>Institutional arrangements:Yes |
| Respondent 6 | Excel/CSV, html, ppt, docx, pdf | Database, Website-URL, email | No URI and<br><br>No linked data | Data License: No<br>Data is open: partially<br>Nature of Policy: institutional<br>Institutional arrangements:Yes |
| Respondent 7 | ppt, docx, pdf, shapefile | Database, flashdrive | No URI and<br><br>No linked data | Data License: No<br>Data is open: partially<br>Nature of Policy: institutional<br>Institutional arrangements:Yes |

## Appendix B. Nature of Participants in the Study

*Appendix B.1. Summary of Respondents that Participated in Interviews*

**Table A3.** Description of respondents in the interview.

| Respondent | Nature of Organization | Data Shared between Stakeholders |
|---|---|---|
| Respondent 1 | Institution interested in food insecurity scenario analysis and early warning based on relevant drivers. | Collects primary data such as price data and volumes of informal trade but relies on a network of partners to access data on farm gain, Climate and weather data from NOAA and USGS, Vegetation conditions, Rainfall data from UNMA, consumer prices, nutrition data, conflict data |
| Respondent 2 | National statistical body | Provides fundamental data sets like administrative boundary data upon which disaster data is overlayed, consumer prices, vulnerability indicators |
| Respondent 3 | Humanitarian that collects open data and maps it on OSM platform | Generates exposure data and maps post disaster impacts. Also relies on data from other institutions such as administration boundaries etc. |
| Respondent 4 | Humanitarian organization for disaster response and preparedness. Collaborates with lead agencies on Flood modeling and early warning, indigenous knowledge EWS, Forecast based financing (FbF) | Generate rapid assessment reports/response data, relies on data from other institutions for joint Hazard, Vulnerability and Risk assessment. Such data include disaster Impact data, land cover, imagery etc |
| Respondent 5 | National Office mandated to coordinate disaster preparedness, prevention and emergency response | Manages DesInventar impact data, Institution does not primarily collect data but rather relies on data from other institutions. Coordinates early warning efforts of other institutions e.g., weather/climate forecasts, food security, crop and pasture conditions, Multi hazard early warning (based on sector data) etc |
| Respondent 6 | Semi autonomous government authority for weather and climate services | Weather data (Rainfall/ humidity/temperature data) and fore casts (daily, 10 day dekadal, monthly and seasonal forecasts), climatic statistics |
| Respondent 7 | Water Resource monitoring department in charge flood hazards | Provide data on water levels in various rivers and lakes being monitored; flood hazard and sanitation data |

*Appendix B.2. Description of Institutions that Participated in Focused Group Discussions*

**Table A4.** Nature of organization where focused group discussions (FGD) participants were drawn.

| No | Nature of Organization |
| --- | --- |
| 1 | Non governmental organization offering relief services to Refugees in Uganda |
| 2 | United Nations agency supporting disaster management in Uganda |
| 3 | Office mandated to coordinate disaster management in Uganda |
| 4 | Higher learning and research institution in Uganda |
| 5 | International Non Governmental Organization that supports humanitarian efforts in Uganda. Also supports data management for community EWS |
| 6 | Ministry in charge of health in Uganda (provides human epidemic data) |
| 7 | Church based humanitarian organization in Uganda |
| 8 | Ministry that provides National data and information on agriculture sector; department involved in food insecurity analysis and early warning |
| 9 | Non governmental Organization involved in climate-smart disaster risk reduction and Forecast based financing |
| 10 | Civil Society Organization helping reduce Community Vulnerability to Climate Change Induced Disasters |
| 11 | Humanitarian Organization spear heading Forecast based Financing in Uganda. |

## Appendix C. Interoperability Patterns

*Technical patterns* include the federated and broadcast patterns. *Semantic patterns* include a list of Linked data patterns, and Ontology design patterns. *Organisational patterns* include the dissemination pattern, rights pattern among others. All Pattern templates and diagrams can be found here: https://github.com/mazimweal/mazimweal.github.io.

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
