# Peer review of "An Empirical Evaluation of Data Interoperability—A Case of the Disaster Management Sector in Uganda"

_ijgi, doi:10.3390/ijgi8110484_

Round 1
Reviewer 1 Report
The paper discusses on a concrete domain of interest the suitability of interoperability measures for data reuse, integration and analysis.
The disaster domain has specific needs and because of emergency, so it is crucial that interoperability has effective and efficient set up services.
The authors identify “interoperability patterns ” bringing solutions to recurrent issues and as means to improve interoperability settings at semantic levels, beyond syntactic interoperability found well established in their example.
The abstract is well written and promising an exiting reading of the rest but the English deteriorates together with the clarity of the content and adequacy with what was said in the abstract. Nonetheless, the study and analysis of the focus groups is certainly worth publishing along with the recommendation for interoperability (with some clarifications). Some of the proposed solutions (if not all) should be argued in their context with a “demonstration” of relevancy.
The current version of the paper is not published and needs a drastic major rewriting: better description of the results and better explanation on how you can derive the solutions (literature is mentioned but this is not sufficient)
As the authors are using Uganda as their example, It would be nice to rapidly list the various types of disasters Uganda has been facing or are managed and by whom.
The involved stakeholders are mentioned in section 3 but a quick set up in the introduction would set the scene. A typical case study (maybe discussed during the FG) could be exposed rapidly to identify the interoperability issues …
In the section “2.1 Disaster Data interoperability”, the authors cite Rezaei et al[17] and Barners-lee[18,19] which are relevant but perhaps
we do not get the feel of the specificities of disaster data challenge for interoperability in the introductory section.
Only the reference [9] is related to disaster and [4] later on.
A short paragraph could be added earlier on in the 2.1 section.
I wondered if in disaster management and hazards management, it is more the entire chain of analysis from re-using existing data, to conflation with newly available data via services using for example SWE (Sensor Web Enablement) principles and standards that one should evaluate instead of just data interoperability via FAIR?
Early Warning System are good examples of what could be assessed too.
For example the section 2 “ Related Work”, some other references on Early Warning System and data interoperability could be added:
Al Hmoudi, A., El Raey, M., & Zeesha, A. (2015). Integrated elements of early warning systems to enhance disaster resilience in the Arab region. Journal of Geodesy and Geomatics Engineering, 2, 73-81.
Samarasundera, E., Hansell, A., Leibovici, D., Horwell, C. J., Anand, S., & Oppenheimer, C. (2014). Geological hazards: From early warning systems to public health toolkits. Health & place, 30, 116-119.
In section “3 Material and Methods”, as proxy of maturity the authors make use of compliance to … to the metrics used in Table A1 but it could be interesting to see the links of these metrics with the FAIR principle for example. It is not clear where do they come from. If these metrics are the production of the authors this should be made clearer (and how they derived them); if not where do they come from?
Because of its importance in the current work the description of the methodology used, i.e. Colpaert [21] could be a bit more expanded. For example what is the ontology / vocabulary used to derive RDF triples. this is discovered in the example … line 132 but would need clarification.
in Section 4.2 the authors talk about emerging patterns but it is not entirely clear where from they do come from in relation to their study … Was this part of the Focus Group’s study?
Section 5.1: Do you mean limitations of the study?
Before starting exploring the various dimension to assess the validity, you could express what is the purpose of this section. The authors reuse the language of validity assessment for measuring instruments (survey questions) used in psychology and public health medicine: self assessment of their evaluation process.
other points … there are more …
line 108 a bit confusing sentence “Interviews from 7 institutions”?
line 140 you mean the number of occurrences? .. and “combined” how?
Table 3 has Table 1 in its title? what is a Revealed pattern in relation to a recommended solution
line 242 typo due to repetition of the first few words
line 261 do you classify satellite imagery as unstructured data?
line 283 we used mixed research
line 284 to 285 … obscure statement!
Author Response
Dear Reviewer,
Thank you for the review comments of our article "An Empirical Evaluation of Disaster Data Interoperability - A Case of Uganda".
These comments have been extremely valuable to improving our paper, as well as stimulating new research agendas.
Based on these comments, we have made the necessary corrections to the manuscript. Attached is the revised manuscript for your reference.
Yours sincerely
Authors
issue No | Issue | response | Action |
1 | The current version of the paper is not published and needs a drastic major rewriting: better description of the results and better explanation on how you can derive the solutions | Agree |
We have re-worked the results, discussion and conclusion sections, In the results section, we indicate how the choice of patterns is dependent on the interoperability assessment In the discussion, we discuss the patterns in relation to a use case-Multi hazard early warning system The conclusion section is adjusted to reflect the changes. The paper was also forwarded to an English language expert for proofreading |
2 |
(literature is mentioned but this is not sufficient) In the section “2.1 Disaster Data interoperability”, the authors cite Rezaei et al[17] and Barners-lee[18,19] which are relevant but perhaps we do not get the feel of the specificities of disaster data challenge for interoperability in the introductory section.
Only the reference [9] is related to disaster and [4] later on. A short paragraph could be added earlier on in the 2.1 section. |
Agree |
Sect 2 has been edited to include a feel of the specificities of disaster data challenge for interoperability |
3 |
As the authors are using Uganda as their example, It would be nice to rapidly list the various types of disasters Uganda has been facing or are managed and by whom. The involved stakeholders are mentioned in section 3 but a quick set up in the introduction would set the scene. A typical case study (maybe discussed during the FG) could be exposed rapidly to identify the interoperability issues … |
Agree |
In section 3, We have added
|
4 | I wondered if in disaster management and hazards management, it is more the entire chain of analysis from re-using existing data, to conflation with newly available data via services using for example SWE (Sensor Web Enablement) principles and standards that one should evaluate instead of just data interoperability via FAIR? | Agree | In this case, we are exploring how patterns can be used to ensure data interoperability. Data, in this case, could be archived/derived, sensor data. Integrated data can already receive as derived data (processed) and otherwise so that processing is done after integration.---this idea has been illustrated while discussing the case of a multi-hazard early warning system in the discussion sector |
5 |
Some of the proposed solutions (if not all) should be argued in their context with a “demonstration” of relevancy. Early Warning System are good examples of what could be assessed too. For example the section 2 “ Related Work”, some other references on Early Warning System and data interoperability could be added: Al Hmoudi, A., El Raey, M., & Zeesha, A. (2015). Integrated elements of early warning systems to enhance disaster resilience in the Arab region. Journal of Geodesy and Geomatics Engineering, 2, 73-81. Samarasundera, E., Hansell, A., Leibovici, D., Horwell, C. J., Anand, S., & Oppenheimer, C. (2014). Geological hazards: From early warning systems to |
Agree |
In this, we study interoperability in the broader disaster sector and derive solutions based on the assessment done. These solutions can be combined to develop different disaster management solutions for instance…. In the discussion section, we demonstrate these solutions can be used in a Multi-hazard Early warning system
Sect 2 has been edited to contextualize the work using a case of Multi-hazard early warning systems |
6 | In section “3 Material and Methods”, as proxy of maturity the authors make use of compliance to … to the metrics used in Table A1 but it could be interesting to see the links of these metrics with the FAIR principle for example. It is not clear where do they come from. If these metrics are the production of the authors this should be made clearer (and how they derived them); if not where do they come from? | Agree | A dedicated table has been included in the Appendix to illustrate how the metrics were derived by the Authors. Corresponding explanations can be found in the methodology section |
7 | Because of its importance in the current work the description of the methodology used, i.e. Colpaert [21] could be a bit more expanded. For example, what is the ontology / vocabulary used to derive RDF triples. this is discovered in the example … line 132 but would need clarification | Agree | We have reworked the methodology section to provide some clarity on the maturity, compatibility tests and pattern identification |
8 | in Section 4.2 the authors talk about emerging patterns but it is not entirely clear where from they do come from in relation to their study … Was this part of the Focus Group’s study? | We have edited the methodology to describe the source of the Global use case. The patterns are derived from challenges and solutions elicited from interviews and FG which are mapped to Global use case patterns (derived from catalogues/literature) while taking into consideration the interoperability assessment. The explanation of choice of patterns is included in the results and discussions sections |
|
9 |
Before starting exploring the various dimension to assess the validity, you could express what is the purpose of this section. The authors reuse the language of validity assessment for measuring instruments (survey questions) used in psychology and public health medicine: self assessment of their evaluation process. line 283 we used mixed research line 284 to 285 … obscure statement! |
Agree | This section was deleted and the corresponding material was incorporated in the methodology and discussion chapters of the paper. |
10 |
line 108 a bit confusing sentence “Interviews from 7 institutions”? |
Agree | edited to make it clearer |
11 |
line 140 you mean the number of occurrences? .. and “combined” how? |
Agree | the section has been reworked |
12 |
Table 3 has Table 1 in its title? what is a Revealed pattern in relation to a recommended solution |
Agre | Changed to more technical terminology “emerging patterns” i.e identified patterns |
13 |
line 242 typo due to repetition of the first few words |
Yes | solved-- |
14 |
line 261 do you classify satellite imagery as unstructured data?
|
No | it is raw data that can be easily transformed using well-established procedures…. This has been corrected too |

Reviewer 2 Report
Let me start by stating that this is one of the most interesting papers I have read in the past few years. I consider myself to be an expert in GIS maturity models and am working on ISO and OGC panels on interoperability standards. I have also worked with the Linked Data community, yet this paper opened doors for me to a completely new set of research.
The first four pages are excellent and require no further edits.
Things slowly start to deteriorate on page 5, where the authors reference the EYE reasoning server and the DesInventar data set without explaining either. I am confident that these are not known to most of the readers of our journal and therefore recommend that the authors expand the contents of this page to two pages.
On page 6, where the authors describe syntactic interoperability results, there is one sentence (lines 165-168) that actually refers to semantics. I recommend that the authors insert a note here referencing section 4.1.4.
Page 7, lines 194-197: I'd hope that after the authors expanded page 5, the readers will now be able to make sense of "focus group discussion 2".
From about page 8 onward, the language deteriorates. It looks as if the authors were rushed and I encourage both a spell and a syntax check of the remaining pages (specially pages 9 and 10). For example, the text should not use general language when the authors really are only talking about disaster management interoperability in Uganda.
I struggle with section 5.1 and if in doubt would just delete the whole section. If, for some reason, it remains, then this section will need some attention. For example, satellite imagery is not unstructured data; instead it is raw data that can be easily transformed using well-established procedures, or in the parlance of the authors, patterns.
On lines 286-87, the sentence "Interview data is collected from seven data points with a theoretical saturation point of three" makes no sense to me whatsoever. What are the authors trying to say here?
Instead of their reasoning expressed in lines 288-89, I would prefer if the authors just told us about what percentage of the Ugandan emergency management community has been covered.
I would prefer if the list of abbreviations on page 10 be sorted alphabetically
Author Response
Dear Reviewer,
Thank you for the review comments of our article "An Empirical Evaluation of Disaster Data Interoperability - A Case of Uganda".
These comments have been extremely valuable to improving our paper, as well as stimulating new research agendas.
Based on these comments, we have made the necessary correction to the manuscript. Attached is the revised paper for reference.
Yours sincerely
Authors
Issue1 : Things slowly start to deteriorate on page 5, where the authors reference the EYE reasoning server and the DesInventar data set without explaining either. I am confident that these are not known to most of the readers of our journal and therefore recommend that the authors expand the contents of this page to two pages
Action 1: We have Reworked the methodology section to provide some clarity on the maturity, compatibility tests and pattern identification
issue2: On page 6, where the authors describe syntactic interoperability results, there is one sentence (lines 165-168) that actually refers to semantics. I recommend that the authors insert a note here referencing section 4.1.4.
Action 2: done
Issue 3: Page 7, lines 194-197: I'd hope that after the authors expanded page 5, the readers will now be able to make sense of "focus group discussion 2".
Action 3:Resolve in Issue No 1
Issue No 4: From about page 8 onward, the language deteriorates. It looks as if the authors were rushed and I encourage both a spell and a syntax check of the remaining pages (specially pages 9 and 10). For example, the text should not use general language when the authors really are only talking about disaster management interoperability in Uganda
Action 4 : we have re-worked the results, discussion and conclusion sections, In the results section, we indicate how the choice of patterns is dependent on the interoperability assessment In the discussion, we discuss the patterns in relation to a use case-Multi hazard early warning system The conclusion section is adjusted to reflect the changes. The paper was also forwarded to an English language expert for proofreading |
Issue 5: I struggle with section 5.1 and if in doubt would just delete the whole section. If, for some reason, it remains, then this section will need some attention. For example, satellite imagery is not unstructured data; instead, it is raw data that can be easily transformed using well-established procedures, or in the parlance of the authors, patterns.
On lines 286-87, the sentence "Interview data is collected from seven data points with a theoretical saturation point of three" makes no sense to me whatsoever. What are the authors trying to say here?
Instead of their reasoning expressed in lines 288-89, I would prefer if the authors just told us about what percentage of the Ugandan emergency management community has been covered.
Action 5:This section was deleted and the corresponding material was incorporated in the methodology and discussion chapters of the paper in a subtle way.
Issue 6: I would prefer if the list of abbreviations on page 10 be sorted alphabetically
Action 6: All abbreviations have been organized in Alphabetical order

Round 2
Reviewer 1 Report
This version has been well improved and would be publishable after consideration of the two last points raided below.
The added /reordering of the text makes it much more readable
1) Perhaps what is still missing is a few sentences concerning the limitations of the study (i.e. based on 16 datasets with 15 institutions and 7 respondents in the FG). How representative these instituions are? the list of dataset used etc…
2) The sentence in the introduction of section 2 “The success of an enhanced early warning system typically depends on the ability of multiple actors to share data/ information (i.e data about past events, real-time streams etc)[12].“ goes in the sense I was suggesting concerning the chain of analysis from data collection to decision-making which is a very important part of Early Warning System, i.e. for efficiency of the decision-making process this chain of transformation and mapping must as smooth as possible and that the role of interoperability. I believe the suggested reference Samarasundera et al. 2014, I mentioned in my previous review is important in that sense in highlighting the role of interoperability standards especially OGC ones in this context.
Author Response
Dear Professor,
Thank you for the review comments of our article "An Empirical Evaluation of Disaster Data Interoperability - A Case of Uganda".
We very much appreciate your contribution towards making this manuscript better.
Based on these comments, we have made the necessary correction to the manuscript. Below is a log with actions taken to solve the issues.
Attached is a pdf showing changes on the previous version of the Manuscript
Yours sincerely
Authors
Issue No | Issue | Comment | Action |
1 | Perhaps what is still missing is a few sentences concerning the limitations of the study (i.e. based on 16 datasets with 15 institutions and 7 respondents in the FG). How representative these instituions are? the list of dataset used etc… | Agree |
We have added a few sentences elaborating how representative the data points are(lines 125-135). A footnote showing the total number of institutions in the sector mailing list has also been added Limitations of data used in the compatibility test have also been mentioned between lines 125-135. The implications for these limitations are discussed on line 227 Tables in the appendix have been edited..to fully explain data points. |
2 | The sentence in the introduction of section 2 “The success of an enhanced early warning system typically depends on the ability of multiple actors to share data/ information (i.e data about past events, real-time streams etc)[12].“ goes in the sense I was suggesting concerning the chain of analysis from data collection to decision-making which is a very important part of Early Warning System, i.e. for efficiency of the decision-making process this chain of transformation and mapping must as smooth as possible and that the role of interoperability. I believe the suggested reference Samarasundera et al. 2014, I mentioned in my previous review is important in that sense in highlighting the role of interoperability standards especially OGC ones in this context. | Agree | We have added the reference by Samarasundera 2014 to support the quoted statement |
